# OpenIFS@home version 1: a citizen science project for ensemble weather and climate forecasting

Sarah Sparrow[1], Andrew Bowery[1], Glenn D. Carver[2], Marcus O. Köhler[2], Pirkka Ollinaho[4], Florian Pappenberger[2], David Wallom[1], Antje Weisheimer[2,3].

[1]Oxford e-Research Centre, Engineering Science, University of Oxford, UK.
[2]European Centre for Medium-Range Weather Forecasts (ECMWF), Reading, UK.
[3]National Centre for Atmospheric Science (NCAS), Atmospheric, Oceanic and Planetary Physics (AOPP), Physics department, University of Oxford, UK.
[4]Finnish Meteorological Institute (FMI), Helsinki, Finland.

*Correspondence to*: Sarah Sparrow (sarah.sparrow@oerc.ox.ac.uk)

**Abstract.** Weather forecasts rely heavily on general circulation models of the atmosphere and other components of the Earth system. National meteorological and hydrological services and intergovernmental organisations, such as the European Centre for Medium-Range Weather Forecasts (ECMWF), provide routine operational forecasts on a range of spatio-temporal scales, by running these models in high resolution on state-of-the-art high-performance computing systems. Such operational forecasts are very demanding in terms of computing resources. To facilitate the use of a weather forecast model for research and training purposes outside the operational environment, ECMWF provides a portable version of its numerical weather forecast model, OpenIFS, for use by universities and other research institutes on their own computing systems.

In this paper, we describe a new project (OpenIFS@home) that combines OpenIFS with a citizen science approach to involve the general public in helping conduct scientific experiments. Volunteers from across the world can run OpenIFS@home on their computers at home and the results of these simulations can be combined into large forecast ensembles. The infrastructure of such distributed computing experiments is based on our experience and expertise with the climateprediction.net and weather@home systems.

In order to validate this first use of OpenIFS in a volunteer computing framework, we present results from ensembles of forecast simulations of tropical cyclone Karl from September 2016, studied during the NAWDEX field campaign. This cyclone underwent extratropical transition and intensified in mid-latitudes to give rise to an intense jet-streak near Scotland and heavy rainfall over Norway. For the validation we use a two thousand member ensemble of OpenIFS run on the OpenIFS@home volunteer framework and a smaller ensemble of the size of operational forecasts using ECMWF's forecast model in 2016 run on the ECMWF supercomputer with the same horizontal resolution as OpenIFS@home. We

present ensemble statistics that illustrate the reliability and accuracy of the OpenIFS@home forecasts as well as discussing the use of large ensembles in the context of forecasting extreme events.

## 1 Introduction

Today there are many ways in which the public can directly participate in scientific research, otherwise known as citizen science. The types of projects on offer range from data collection/generation, for example taking direct observations at a particular location such as in the British Trust for Ornithology's "Garden BirdWatch" (RSPB Garden BirdWatch), through data analysis, such as image classification in projects such as Zooniverse's galaxy classification (Simpson et al 2014) and finally data processing. This final class of citizen science includes those projects where citizens donate time on their computer to execute project applications. Examples of this class of citizen science project are known as volunteer or crowd computing applications. There is an extremely wide variety of different projects making use of this paradigm, the most well-known of which is searching for extra-terrestrial life with SETI@home (Sullivan III et al., 1997). Projects of this type are underpinned by the Berkeley Open Infrastructure for Network Computing (BOINC, Anderson, 2004) that distributes simulations to the personal computers of their public volunteers who have donated their spare computing resources.

For over 15 years one such BOINC based project, climate*prediction*.net (CPDN) has been harnessing public computing power to allow the execution of large ensembles of climate simulations to answer questions on uncertainty which would otherwise not be possible to study using traditional High Performance Computing (HPC) techniques (Allen, 1999; Stainforth et al., 2005). Volunteers can sign-up to CPDN through the project website and are engaged and retained through the mechanisms detailed in (Christensen et al., 2005). As well as facilitating large ensemble climate simulations the project has also increased public awareness of climate change related issues. Through the CPDN platform, volunteers are notified of the scientific output that they have contributed towards (complete with links to the academic publication) and through the project forums and message boards can engage directly with scientists about the experiments being undertaken. Public awareness is also raised by press coverage of the project (e.g. Gadgets that give back: awesome eco-innovations, from Turing Trust computers to the first sustainable phone or Climate Now | Five ways you can become a citizen scientist and help save the planet), scientific outputs (e.g. 'weather@home' offers precise new insights into climate change in the West; How your computer could reveal what's driving record rain and heat in Australia and NZ; Looking, quickly, for the fingerprints of climate change) and through live experiments undertaken directly with media outlets such as The Guardian (Schaller et al., 2016) and British Broadcasting Corporation (BBC, Rowlands et al., 2012). To date the analysis performed by CPDN scientists and volunteers can be broadly classified into three different themes. The first is climate sensitivity analysis where plausible ranges of climate sensitivity are mapped through generating large, perturbed parameter ensembles (e.g. Millar et al., 2015; Rowlands et al., 2012; Sparrow et al., 2018b; Stainforth et al., 2005; Yamazaki et al., 2013). The second is simulation bias reduction methods through perturbed parameter studies (e.g. Hawkins et al., 2019; Li et al., 2019; Mulholland et al., 2017). The third category is extreme weather event attribution studies where quantitative assessments are made

of the change in likelihood of extreme weather events occurring between past, present and possible future climates (e.g. Li et al., 2020; Otto et al., 2012; Philip et al., 2019; Rupp et al., 2015; Schaller et al., 2016; Sparrow et al., 2018a).

To increase confidence in the outcomes of large ensemble studies it is desirable to compare results across multiple different models. Whilst large (order 100 member) ensembles can be, and are, produced by individual modelling centres, this requires a great deal of coordination across the community on experimental design and output variables.  The computing resources required to produce very large (>10,000 member) ensembles are not readily available outside of citizen science projects such as CPDN. Therefore, enabling new models to work within this infrastructure to address questions such as those outlined above is very desirable.

In this paper we detail the deployment of the European Centre for Medium-Range Weather Forecasts (ECMWF) OpenIFS model within the CPDN infrastructure as the OpenIFS@home application. This new facility enables the execution of ensembles of weather forecast simulations (ranging from 1 to 10,000+ members) at scientifically relevant resolutions to:

- Study the predictability of forecasts especially for high impact extreme events.
- Explore interesting past weather and climate events by testing sensitivities to physical parameter choices in the model.
- Help the study of probabilistic forecasts in a chaotic atmospheric flow and reduce uncertainties due to nonlinear interactions.
- Support the deployment of current experiments performed with OpenIFS to run in OpenIFS@home provided certain resource constraints are met.

## 100  2. The ECMWF OpenIFS model

The OpenIFS activity at ECMWF began in 2011, with the objective of enabling the scientific community to use the ECMWF Integrated Forecast System (IFS) operational numerical weather prediction model in their own institutes for research and education. OpenIFS@home as described in this paper uses the OpenIFS release based on IFS cycle 40 release 1, the ECMWF operational model from November 2013 to May 2015. The OpenIFS model differs from IFS as the data assimilation and observation processing parts are removed from the OpenIFS model code. The forecast capability of the two models is identical however, and the OpenIFS model supports ensemble forecasts and all resolutions up to the operational resolution. OpenIFS consists of a spectral dynamical core, comprehensive set of physical parametrizations, surface model (HTESSEL) and ocean wave model (WAM). A more detailed description of OpenIFS can be found in Appendix A. The relative contribution of model improvements, reduction in initial state error and increased use of observations to the IFS forecast performance is discussed in detail in Magnusson and Källén, (2013). A detailed scientific and technical description of IFS can be found in open access scientific manuals available from the ECMWF website (ECMWF, 2014a,b,c,d).

### 3. OpenIFS@home BOINC application

#### 3.1 Technical requirements and challenges

When creating a new volunteer computing project, there are a number of requirements for both the science team developing it and the citizen volunteers that will execute it. As such they can be considered boundary constraints. These are listed below;

1) The model used to build the BOINC application should be unchanged. There are two main advantages from this. First the model itself does not require extensive revalidation, second if errors are found within the BOINC based model an identically configured non-BOINC version may be executed locally for diagnostic purposes. As OpenIFS is currently designed for simulation on unix/linux systems, initial development of OpenIFS@home has also been limited to this platform, thereby preventing the need for a detailed revalidation. Consequently OpenIFS@home is currently limited to the Linux CPDN volunteer population, around 10% of the 10,000 active volunteers registered with CPDN.

2) The model configuration for an experiment and the formulation of initial conditions and ancillary files should remain unchanged from that used in a standard OpenIFS execution to allow easy support by the OpenIFS team in ECMWF and debugging by the CPDN.

3) Configuration of the ensembles should be simple, requiring minimal changes to input files to launch a large batch of simulations. Web forms developed for this minimise the possibility of error in the configuration.

4) Model performance, when running on volunteers' systems, should be acceptable such that results are produced at a useful frequency for the submitting researcher, but also that the time to completion of an individual simulation workunit is practical for the volunteers' systems. This dictates the resolution of the simulation that can be run; a lower resolution than that utilised operationally, but still scientifically useful.

5) The model must not generate excessive volumes of output data such that volunteers' network connections are overwhelmed. This requires integration of existing measures to analyse the model configuration so that the CPDN team can validate the expected data volumes before submission.

6) The model binary executable needs to minimise dependencies on the specific configuration of the system found on the volunteer computers. Therefore, the compilation environment for OpenIFS@home needs to use statically linked libraries wherever possible, distributing these in a single application package.

#### 3.2 Porting OpenIFS to a BOINC environment

To optimise OpenIFS for execution within BOINC on volunteer systems there are a number of changes that are required to the model beyond setup and configuration changes. The majority of these may be classified in terms of understanding and restricting the application footprint in terms of both overall size and resource usage during execution.

OpenIFS is designed to work efficiently across a range of computing systems, from massively parallel high-performance computing systems to a single multi-core desktop. As BOINC operates optimally if each application execution is restricted to a single core on a client system, understanding and reducing memory usage becomes a priority, determining possible resolutions the model can be executed at. During the initial application development, a spectral resolution of T159, equivalent to a grid spacing of approximately 125km (see Appendix for more details of the model's grid structure) with 60 vertical level was chosen to ensure execution would complete within one or two days whilst still maintaining satisfactory scientific performance. Typical CPDN simulations run for considerably longer allowing flexibility in future utilisation of this application.

Since OpenIFS@home will run on a single compute core, the MPI (message passing) parallel library was removed from the OpenIFS code, though the ability to use OpenMP was retained for possible future use. This reduces the memory footprint and size of the binary executable.

A model restart capability is necessary as the volunteer computer may be shutdown at any point in the execution. OpenIFS provides a configurable way of enabling exact restarts, with an option to delete older restart files.   This was added to the model configuration to prevent excessive disk use on the volunteer's computers.

There is also the requirement to transfer to volunteer systems the configuration files that control the execution of the model and the return of model output files. The design of OpenIFS makes it inherently suitable for deployment under BOINC.  Input and output files use the standard GRIB format (World Meteorological Organisation (WMO), 2003) that was originally designed for transmission over slow telecommunication lines.  The model output files are separated into spectral and gridpoint fields. Each model level of each field is encoded in a self-describing format whilst the field data itself is packed into a specified, 'lossy', bit-precision. This greatly reduces the amount of data transmission whilst the self-describing nature of each of the GRIB fields supports a 'trickle' of output results as the model runs. Scientists are expected to carefully choose the model fields, and levels, required to minimise output file sizes and transmission times to the CPDN servers. This is an optimisation exercise which is supported by CPDN, with exact thresholds depending on frequency of return as well as absolute file size due to differences in volunteer's internet connectivity.

The GRIB-1&2 definitions do, however, introduce one difficulty. The encoding of the ensemble member number only supports values up to 255. To overcome this, custom changes were made to the output GRIB files to allow exploitation of the much larger ensembles that could be distributed within OpenIFS@home. Specifically, four spare bytes in the output gridpoint GRIB fields were used to create a custom ensemble perturbation number (defined in local part of section 1 in GRIB-1 output; section 3 of GRIB-2 output). The custom GRIB templates must be distributed with the model to the volunteer's computer and subsequently used when decoding the returned GRIB output files.

## 4. Demonstration

### 4.1 Case study: Storm Karl

Recent research into mid-latitude weather predictability has focused on the role of diabatic processes. Research flight campaigns provide in-situ measurements of diabatic and other physical processes against which models can be validated. The NAWDEX flight campaign (Schäfler et al., 2018) focused on weather features associated with forecast errors, for example the poleward recurving of tropical cyclones which is known to be associated with low predictability (Harr et al., 2008). To demonstrate the
new OpenIFS@home facility, we simulated the later development of a tropical cyclone (TC) in the North Atlantic that occurred during the NAWDEX campaign. In September 2016 TC Karl underwent extratropical transition and its path moved far into the midlatitudes. The storm resulted in high impact weather in north-western Europe (Euler et al., 2019). After leaving the subtropics on 25 September ex-TC Karl moved northwards and merged with a weak pre-existing cyclone. This resulted in rapid
intensification and the formation of an unusually strong jet streak downstream near Scotland two days later. This initiated further development with heavy and persistent rainfall over western Norway.

### 4.2 Experimental set-up and initial conditions

A 6-day forecast experiment was designed to capture the extratropical transition of TC Karl and the associated high impact weather north of Scotland and near the Norwegian west coast. The forecasts
were initialised on 25 September 2016 at 00:00 UTC (see Fig.1). The wave model was switched off in OpenIFS for this experiment. Compared to the operational forecasting system at ECMWF and other major weather centres the OpenIFS grid resolution used here is coarse (~125km grid spacing) and hence the model's ability to resolve orographic effects over smaller scales will be limited.

To represent the uncertainty in the initial conditions and to evaluate the range of possible forecasts, a 2,000-member ensemble with perturbed initial conditions was launched. The ECMWF data assimilation system was used to create 250 perturbed initial states.  Each of these 250 states was then used for 8 forecasts in the 2,000 member ensemble. A different forecast realisation for each set of 8 forecasts was generated by enabling the stochastic noise in the OpenIFS physical parametrizations.
The initial state perturbations in the ECMWF operational IFS ensemble are generated by combining so-called Singular Vectors (SV) with an Ensemble of Data-Assimilations (EDA) (Buizza et al., 2008; Isaksen et al., 2010; Lang et al., 2015). The SVs represent atmospheric modes which grow rapidly when perturbed from the default state. In the operational IFS ensemble, the modes which result in maximum
total energy deviations in a 48h forecast lead time are targeted. Fifty of these modes are searched for in the Northern Hemisphere, fifty in the Southern Hemisphere, and 5 modes per active Tropical Cyclone in the Tropics. The final SV initial state perturbation fields are constructed as a linear combination of the found SVs (Leutbecher and Palmer, 2008). The EDA-based perturbations on the other hand, try to assess uncertainties in the observations (and the model itself) used in the Data Assimilation (DA). This
is achieved by running the IFS DA at a lower resolution multiple times and applying perturbations to

the used observations and to the model physics. In the operational IFS ensemble, 50 of these DA cycles are run (Lang et al., 2019). The final perturbations fields that the operational IFS ensemble uses are a combination of both of the perturbed fields. Here, we apply the same methodology as in the operational IFS ensemble initialisation. The only differences are that (i) the used model version and resolution

differ from the operational setup, (ii) only 25 DA cycles are run with a +/- symmetry to construct 50 initial states, and (iii) we calculate 250 SV modes in the extra-tropics, instead of the default 50. This was motivated by the discussion in (Leutbecher and Lang, 2014). The Stochastically Perturbed Parameterization Tendencies (SPPT) scheme (Buizza et al., 2007; Palmer et al., 2009; Shutts et al., 2011) was used on top of the initial state perturbations to represent model error in the ensemble.


The results from this experiment were compared against output from an ensemble of the same size as ECMWF's operational forecasts (51 members) run at the same horizontal resolution as our forecast experiment using the current operational IFS cycle at the time of writing (CY46R1).

## 4.3 Performance

### 4.3.1 BOINC application performance

Figure 2a shows the behaviour of the batch of simulations (OpenIFS dashboard, 2019), detailing how long simulations were in the queue (yellow), took to run (purple), and took to accumulate an ensemble of successful results (blue). The overall percentage of successfully completed runs in the batch compared to those distributed is also shown in the title. Medians rather than means are quoted as distributions can have long tails. This is due to the nature of the computing resources used, where for a

variety of reasons a small number of simulations (workunits) may go to systems that may not run or connect to the internet for a non-trivial period after receiving work. This validation batch was run on the CPDN development site where fewer systems are connected than on the main site, but they are more likely to be running continuously. The median queue time was 45.64 hours and the 6 day simulations

had a median runtime of 3.25 hours across the difference volunteer machines. Half of the batch (i.e. 50% of the ensemble) was returned after 51.76 hours with 80% completion (the criteria typically chosen for closure of a batch) being achieved after 80.68 hours. The median run time distribution (purple) shows a bi-modal structure which reflects the different system specifications and project connectivity of client machines connected to the development site. As detailed in Anderson 2004 and Christensen et al.,

2005, each volunteer can configure their own project connectivity, available resource and specify during which times their system can be used to compute work so these timings should be viewed as indicative rather than definitive. Figure 2b shows how the run time from a representative batch of OpenIFS@home simulations compares to the other UK Met Office model configurations available on the CPDN platform (although typically these are used to address different questions) and demonstrates that not

only is the OpenIFS@home run time comparable to the different embedded regional models in weather@home, it is also among the faster running models on the platform. The OpenIFS@home application running at this resolution (T159L60) requires 3.2Gb of storage and 5.37Gb of Random Access Memory (RAM).

Although individual simulations will not necessarily be bit reproducible when run on systems with different operating systems and processor types. Knight et al., 2007 demonstrate the effect of hardware and software is small relative to the effect of parameter variation and can be considered equivalent to those differences caused by changes in initial conditions. Given the large ensembles involved, the properties of the distributions themselves are not expected to be affected by different mixes of hardware
in computing individual ensemble members.

### 4.3.2 Meteorological performance

Storm Karl, as it moved eastward across the North Atlantic, was associated with a band of low surface pressure that reached over a region of Northern Scotland 60 hours into the forecasts and near Bergen at the coast of Norway at 72 hours (see green and orange boxes in Fig 3 for the regions).
We discuss here the results of the OpenIFS@home forecast using 2,000 ensemble members run at a horizontal resolution of approx. 125 km (the T159 spectral resolution). These forecasts will be contrasted with two 51-ensemble member forecasts of the IFS run on the ECMWF supercomputer: a low-resolution experiment also at 125 km (T159) resolution and the operational forecast at the time of
Storm Karl which has a resolution of approx. 18 km (T1279), that is almost an order of magnitude finer. ECMWF's operational weather forecasts comprise of 51 individual ensemble members and serves here as a benchmark.

The OpenIFS@home ensemble predicted a distribution of surface pressure averaged over the Northern
Scotland area with a mean of approx. 1011 hPa and a long tail towards low pressure values (Fig 4a). The analysis value of 1001 hPa is just at the lowest edge of the distribution indicating that while the OpenIFS@home model was able to assign a non-zero probability to this extreme outcome, it did not indicate a seriously large risk for such small values. In comparison, the forecast with the standard operational prediction ensemble size of 51 members (Fig 4b) did not even include the observed
minimum in its tails, implying that the observed event was virtually impossible to occur. This clearly demonstrates the power of our large ensemble which, while not assigning a significant probability to the observed outcome, did include it as a possible though unlikely outcome. The overestimation of the surface pressure in the OpenIFS@home forecasts is hardly surprising because the magnitude of pressure minima strongly depends on the horizontal model resolution. For example, the operational high-
resolution ECMWF forecast is shown in Fig 4c. The distribution is nearly uniformly distributed between 999 hPa and 1012 hPa. The analysis value lies well within that range, interestingly though at a local minimum of the distribution. With the application of suitable calibration, or adjustment, for the horizontal model resolution-dependent underestimation bias in the surface pressure mean, the example of storm Karl demonstrates the power of large ensembles to assign non-zero probabilities to extreme
outcomes at the very tails of the distribution

The precipitation forecasts for Northern Scotland are shown in panels d)-f) of Fig 4. OpenIFS@home forecasts a substantial probability to the possibilities of rainfall values larger than the analysis. A traditional-sized ensemble of the same horizontal resolution considers the observed outcome much less

likely than the large OpenIFS@home ensemble. The high-resolution forecast at operational resolution did arguably not perform much better than OpenIFS@home even though the low-pressure system itself would be better simulated

The forecasts of surface pressure and precipitation over the region near Bergen is shown in Fig 5. Similar to the performance north of Scotland, the large ensemble of OpenIFS@home (a) does include in its distribution the observed low pressure value while in the case of a 51-member IFS T159 ensemble even the lowest forecast value was above the analysis (b). The high-resolution operational IFS forecast (c) gave a higher probability to the observed outcome than OpenIFS@home but also considered it extreme within its predicted range.

The extreme precipitation amount of nearly 20 mm/day in the analysis for the region around Bergen was only captured by the high-resolution operational IFS forecast (f) which is likely a result of the much improved representation of the small-scale orography over the coast of Norway in runs with high horizontal resolution, with implications for orographic rain amounts. While the entire distribution of the 51-member IFS ensemble was far off the observed amount without any indication of possibly more extreme outcomes (e), the large ensemble of OpenIFS@home (d) produced a long tail towards extreme precipitation amounts which nearly reached 20mm/day.

The forecast accuracy of the extreme meteorological conditions of storm Karl is influenced by 3 key factors: i) a good physical model that can simulate the atmospheric flow in highly baroclinic extratropical conditions as an extratropical low-pressure system; ii) higher horizontal resolution allows a better resolution of the storm, and iii) a large ensemble that samples a wide range of uncertainties given the simulated flow for a given resolution. OpenIFS@home is built on the world-leading NWP forecast model of ECMWF which enables our distributed forecasting system to use the most advanced science of weather prediction. Arguably, a storm like Karl will be better resolved with higher horizontal resolution, as becomes clear in our demonstration of the IFS performance in two contrasting resolutions. However, there are other meteorological phenomena where horizontal resolution does not play a similarly large role in the successful prediction of extreme events. For these situations, the availability of very large ensembles that enable a meaningful sampling of the tails of the distribution and with it the risks for extreme outcomes, will be most valuable. Our storm Karl analysis has made that point very clear by showing a substantial improvement in the probabilistic forecasts of both very low surface pressure (and associated winds) and large rainfall totals.

## 5. Conclusions

This paper introduced the OpenIFS@home project (version 1) that enables the production of very large ensemble weather forecasts, supporting types of studies previously too computationally expensive to attempt and growing the research community able to access OpenIFS. This was completed with the help of citizen scientists who volunteered computational resources and the deployment of the ECMWF OpenIFS model within the CPDN infrastructure as the OpenIFS@home application. The work is based

on the climate*prediction*.net and weather@home systems enabling a simpler and more sustainable deployment.

We validated the first use of OpenIFS@home in a volunteer computing framework for ensemble forecast simulations using the example of storm Karl (September 2016) over the North Atlantic. Forecasts with 2000 ensemble members were generated for 6 days ahead and computed by volunteers within ~3 days. Significantly smoother probability distribution can be created than forecasts generated with significantly less ensemble members. In addition, the very large ensemble can represent the uncertainty better in particular in the tails of the forecast distributions, allowing higher accuracy of the probability of extremes of the forecast distribution. The relatively low horizontal resolution of OpenIFS@home when compared with typical operational NWP resolutions and the potential implications due to the resolution are, however, a limitation that always must be kept in mind for specific applications. This system has significant future potential and offers opportunities to address topical scientific questions such as:

- Comprehensive sensitivity analyses to attribute sources of uncertainty, which dominate the meteorological forecasts and meteorological analyses directing where to allocate resources for future research i.e. understanding how much meteorological uncertainty is generated through the land surface parameterisation in comparison to the ocean.

- Investigating the tails of distribution and forecast outliers which are important for risk-based decision making in particular in high impact low probability scenarios e.g. Tropical cyclone landfall.

- Improve understanding of non-linear interactions of all earth system components and their uncertainties will provide valuable insight into fundamental model processes. Not only will large initial conditions ensembles be possible, but also large multi-model perturbed parameter experiments.

We have demonstrated that the current application as deployed produces scientifically relevant results within a useful timeframe, whilst utilising acceptable amounts of computational resources on volunteering citizen scientist's personal computers. However, further developments have been identified as desirable in the future use of the facility.  For instance, developing a working application for Windows (and MacOS) systems would significantly increase the number of volunteers available to compute OpenIFS@home simulations, which in turn would result in a reduction in queue time for simulations as well as engaging public volunteers from a wider community. Another possible future development will look at utilizing multiple cores via the OpenMP multi-threading capability of OpenIFS. As new versions of the OpenIFS model are released, the OpenIFS@Home facility will be updated as resources allow.

In terms of  potential areas of future scientific use of openIFS@home, research on understanding and predicting compound extreme events (for example, a heat wave in conjunction with a meteorological and hydrological drought) will be of interest. ECMWF's operation ensemble size of 51 members makes such investigations very difficult and limited in their scope, while the very large ensemble set-up of openIFS@home provides an ideal framework for the required sample sizes of multi-variate studies. We are planning to use the system for predictability research on a range of time scales from days to weeks and months, with potential idealised climate applications also feasible in the longer term.

## Appendix A: OpenIFS description

OpenIFS uses a hydrostatic dynamical core for all forecast resolutions, with prognostic equations for the horizontal wind components (vorticity and divergence), temperature, water vapour and surface pressure. The hydrostatic, shallow-atmosphere approximation, primitive equations are solved using a two-time-level, semi-implicit semi-Lagrangian formulation (Hortal, 2002; Ritchie et al., 1995; Staniforth and Côté, 1991; Temperton et al., 2001). OpenIFS is a global model and does not have the capability for

limited-area forecasts.

The dynamical core is based on the spectral transform method (Orszag, 1970; Temperton, 1991). Fast-Fourier Transforms (FFTs) in the zonal direction and Legendre transforms (LT) in the meridional direction are used to transform the representation of variables to and from grid-point to spectral space.

The spectral representation is used to; compute horizontal derivatives, efficiently solve the Helmholtz equation associated with the semi-implicit time-stepping scheme and apply horizontal diffusion. The computation of semi-Lagrangian horizontal advection, the physical parameterizations and the non-linear right-hand side terms are all computed in grid-point space. The horizontal resolution is therefore represented by both the spectral truncation wavenumber (the number of retained waves in spectral

space) and the resolution of the associated Gaussian grid. Gaussian grids are regular in longitude but slightly irregular in latitude with no polar points. Model resolutions are usually described using a Txxx notation where xxx is the number of retained waves in spectral representation. In the vertical a hybrid sigma pressure-based coordinate is used, in which the lowest layers are pure so-called 'sigma' levels, whilst the topmost model levels are pure pressure levels (Simmons and Burridge, 1981). The vertical

resolution varies smoothly with geometric height, finest in the planetary boundary layer becoming coarser towards the model top. A finite element scheme is used for the vertical discretization (Untch and Hortal, 2004). In this paper, all OpenIFS@home forecasts used the T159 horizontal resolution on a linear model grid with 60 vertical levels. This approximates to a resolution of 125km at the equator, or 'N80' grid.


The OpenIFS model includes a comprehensive set of sub-grid parametrizations representing radiative transfer, convection, clouds, surface exchange, turbulent mixing, sub-grid-scale orographic drag and non-orographic gravity wave drag. The radiation scheme uses the Rapid Radiation Transfer Model (RRTM) (Mlawer et al., 1997) with cloud radiation interactions using the Monte Carlo Independent

Column Approximation (McICA) (Morcrette et al., 2008). Radiation calculations of short- and long-wave radiative fluxes are done less frequently than the timestep of the model and on a coarser grid. This is relevant for implementation in the BOINC framework because this calculation of the fluxes represents the high-water memory usage of the model. The moist convection scheme uses a mass-flux approach representing deep, shallow and mid-level convection (Bechtold et al., 2008; Tiedtke, 1989)

with a recent update to the convective closure for significant improvements in the convective diurnal cycle (Bechtold et al., 2014). The cloud scheme is based on Tiedtke (1993) but with an enhanced representation of mixed-phase clouds and prognostic precipitation (Forbes et al., 2011a, 2011b). The HTESSEL tiled surface scheme represents the surface fluxes of energy, water and corresponding sub-surface quantities. Surface sub-grid types of vegetation, bare soil, snow and open water are represented

(Balsamo et al., 2009). Unresolved orographic effects are parametrized according to Beljaars et al. (2004) and Lott and Miller (1997). Non-orographic gravity waves are parametrized according to Orr et al. (2010). The sea-surface has a two-way coupling to the ECMWF wave model (Janssen, 2004). Monthly mean climatologies for aerosols, long-lived trace gases, surface fields such as sea-surface temperature are read from external fields provided with the model package. Although IFS includes an
ocean model for operational forecasts, OpenIFS does not include it.

In order to represent random model error due to unresolved subgrid-scale processes, OpenIFS includes the stochastic parameterization schemes of IFS (see Leutbecher et al. (2017) for an overview). For example, the SPPT scheme perturbs the total tendencies from all physical parameterizations using a
multiplicative noise term (Buizza et al., 2007; Palmer et al., 2009; Shutts G et al., 2011).

## Code availability

The BOINC implementation of OpenIFS, as distributed by CPDN, includes a free personal binary-only license to use the custom OpenIFS binary executable on the volunteer computer. Researchers who need to modify the OpenIFS source code for use in OpenIFS@home must have an OpenIFS software source
code license.

A software licensing agreement with ECMWF is required to access the OpenIFS source distribution: despite the name it is not provided under any form of open source software license. License agreements are free, limited to non-commercial use, forbid any real-time forecasting and must be signed by research
or educational organisations. Personal licenses are not provided. OpenIFS cannot be used to produce nor disseminate real-time forecast products. ECMWF has limited resources to provide support, so may temporarily cease issuing new licenses if deemed difficult to provide a satisfactory level of support. Provision of an OpenIFS software license does not include access to ECMWF computers nor data archive other than public datasets.
OpenIFS requires a version of the ECMWF ecCodes GRIB library for input and output, version 2.7.3 was used in this paper (though results are not dependent on the version). All required ecCodes files, such as the modified GRIB templates, are included in the application tarfile available from Centre for Environmental Data Analysis (http://www.ceda.ac.uk, see data availability section below for details).
Version 2.7.3 of ecCodes can also be downloaded from the ECMWF github repository (https://github.com/ecmwf), though note the modified GRIB templates included in the application tarfile must be used.

Parties interested in modifying the model source code should contact ECMWF by emailing openifs-
support@ecmwf.int, to request a license outlining their proposed use of the model. Consideration may be given to requests that are judged to be beneficial for future ECMWF scientific research plans, or from scientists involved in new or existing collaborations involving ECMWF. See the following webpage for more details: https://software.ecmwf.int/oifs.

All bespoke code that has been produced in the creation of OpenIFS@home is kept in a set of publicly available open source GitHub repositories under the CPDN-git organisation (https://github.com/CPDN-git). The exact release versions (1.0.0) are archived on Zenodo (Bowery and Carver 2020; Sparrow 2020a, Sparrow 2020b, Sparrow 2020c, Uhe and Sparrow 2020).

The OpenIFS@home binary application code version 2.19 together with the post-processing and plotting scripts used to analyse and produce the figures in this paper are included within the deposit at the CEDA data archive (details provided in the data availability section).

## Data availability

The initial conditions used for the Tropical Cyclone Karl forecasts described in this paper together with
the full set of model output data for the experiment used in this study is freely available (Sparrow et al 2021) at the Centre for Environmental Data Analysis (http://www.ceda.ac.uk).

## Author contribution

SS was instrumental in specifying overall concept as well as the BOINC application design for OpenIFS@home, developed web interfaces for generating ensembles, managing ancillary data files and
monitoring distribution of ensembles along with writing training documentation. AB developed the BOINC application for OpenIFS@home which is distributed to client machines and wrote scripts for submission of OpenIFS@home ensembles into the CPDN system. GDC developed the BOINC version of OpenIFS deployed in OpenIFS@home. MOK contributed to the preparation of the experiment initial conditions. PO generated the perturbed initial states. DW was influential in specifying the overall
concept as well as the specific BOINC application design of OpenIFS@home. FP was influential in the development of OpenIFS@home. AW was instrumental in the conceptual idea of using OpenIFS as a state-of-the-art weather prediction model for citizen science large ensemble simulations. AW also developed some of the diagnostics. All authors contributed to the writing of the manuscript.

**Competing interests**
The authors declare that they have no conflict of interest.

**Acknowledgements**
We gratefully acknowledge the personal computing time given by the CPDN moderators for this
project. We are grateful for the assistance provided by ECMWF for solving the GRIB encoding issue for very large ensemble member numbers.

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

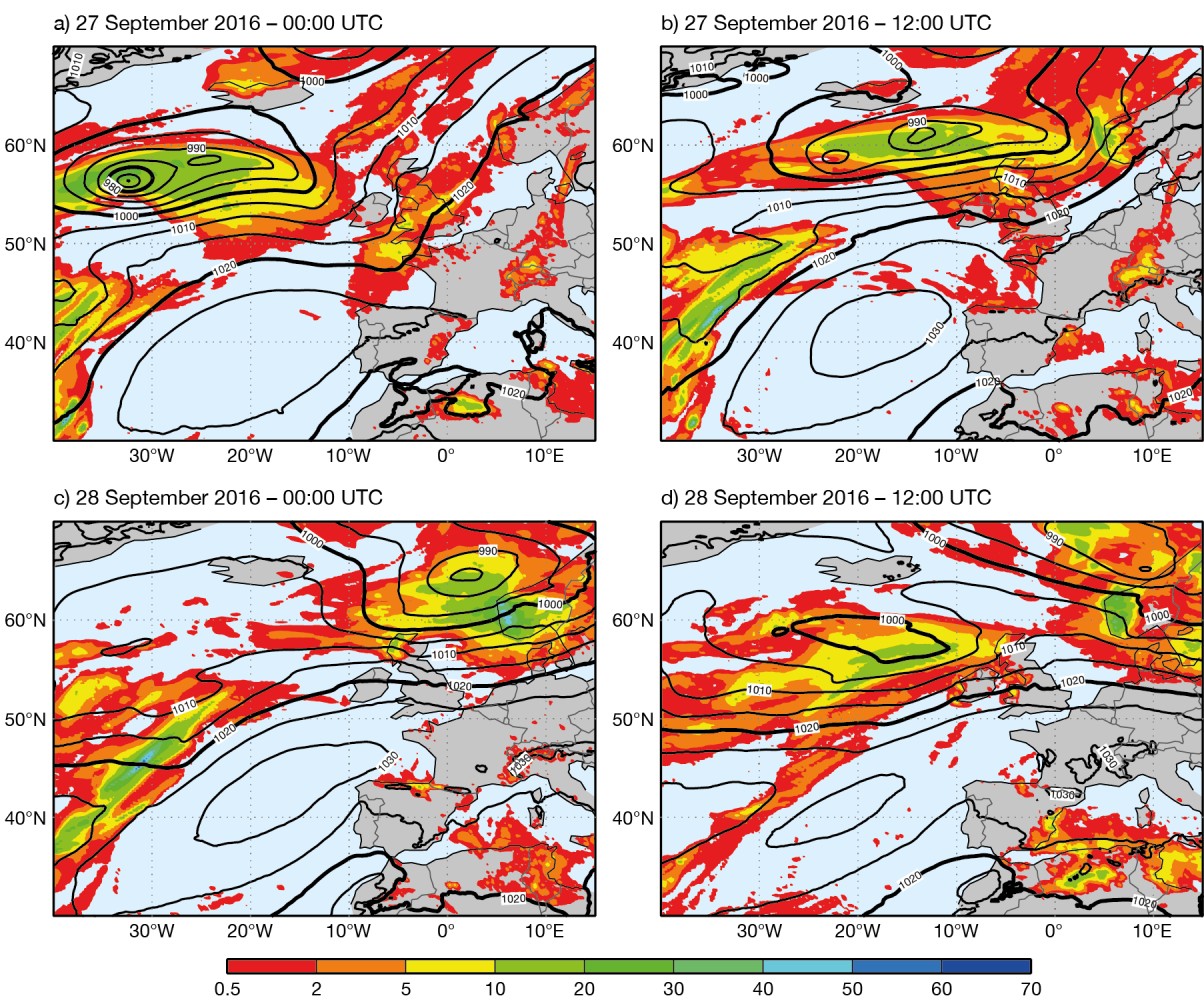

**Figure 1: Evolution of storm Karl on 27th and 28th September 2016 showing the downstream impact with heavy rainfall over west Norway. Contour lines display mean sea level pressure (hPa) from the ECMWF operational analysis. Colour shading shows the 12-hourly accumulated total precipitation (mm) from the ECMWF operational forecast.**


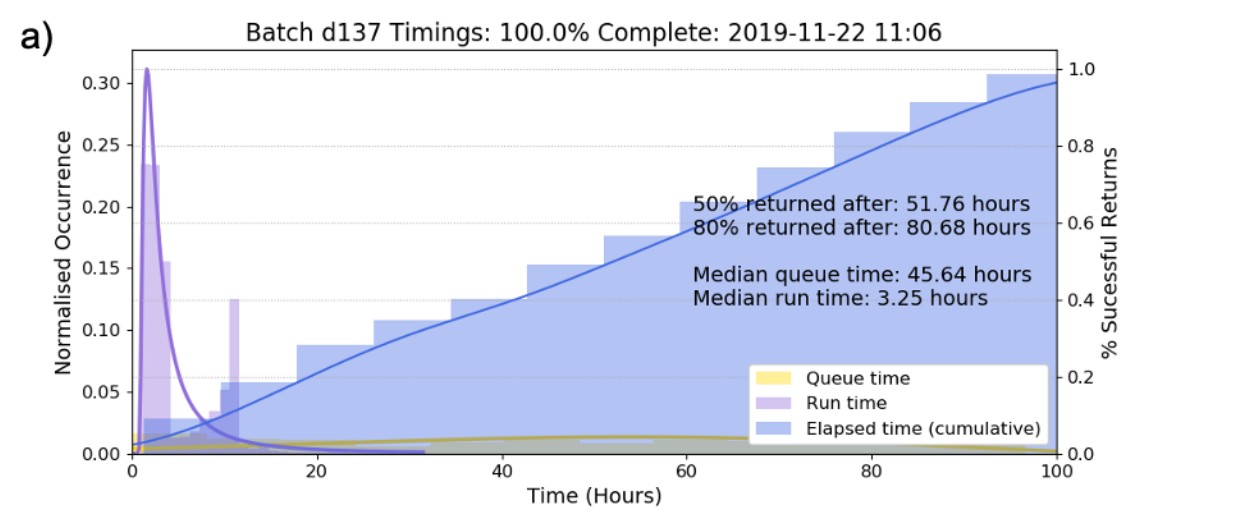

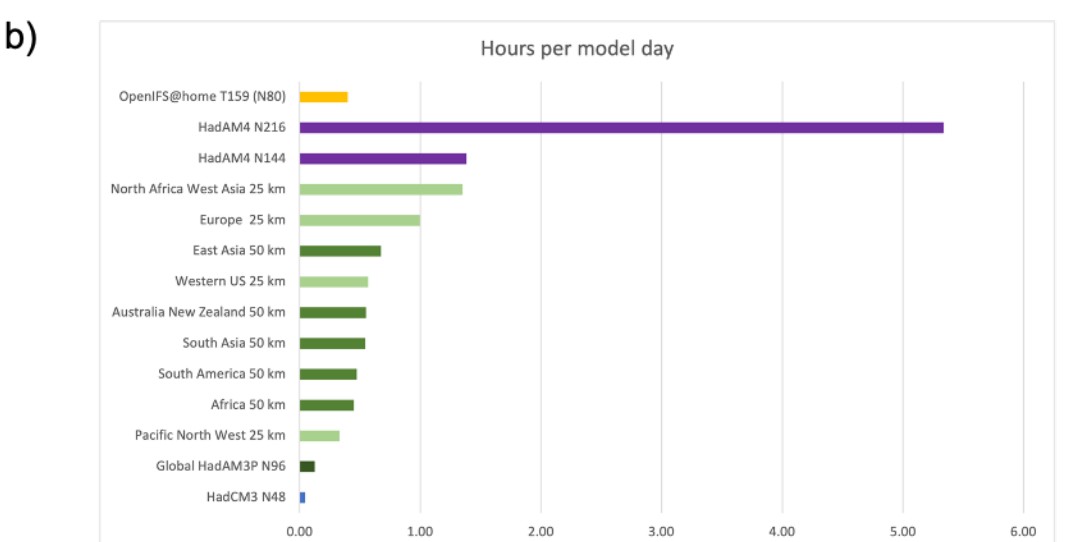

Figure 2: (a) The relevant timings associated with the validation batch. The queue time (yellow) and run time (purple) distributions are straight occurrence distributions whereas the elapsed time (blue) is a cumulative distribution expressed as a percentage of the successful returns. (b) Run time information in hours per model day based on a representative batch for applications on the CPDN platform (note these numbers are indicative rather than definitive). ECMWF OpenIFS@home is depicted in yellow, UK Met Office weather@home (HadAM3P with various HadRM3P regions) configurations in green (with light green indicating a 25km embedded region, mid green a 50km embedded region and dark green where only the global driving model is computed). The UK Met Office low resolution coupled atmosphere-ocean model HadCM3 is shown in blue and the high resolution global atmosphere HadAM4 at N144 (~90km mid-latitudes) and N216 (~60km mid-latitudes) are shown in purple.

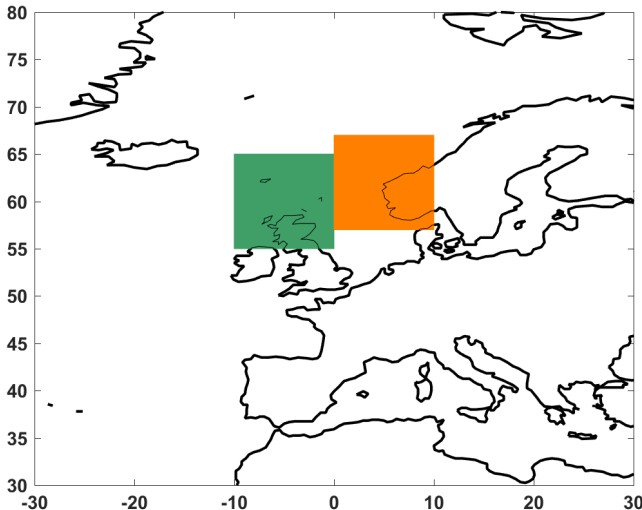

**Figure 3: Two regions over Northern Scotland (green) and around Bergen (orange) that have been used in the diagnostics of the model performance.**

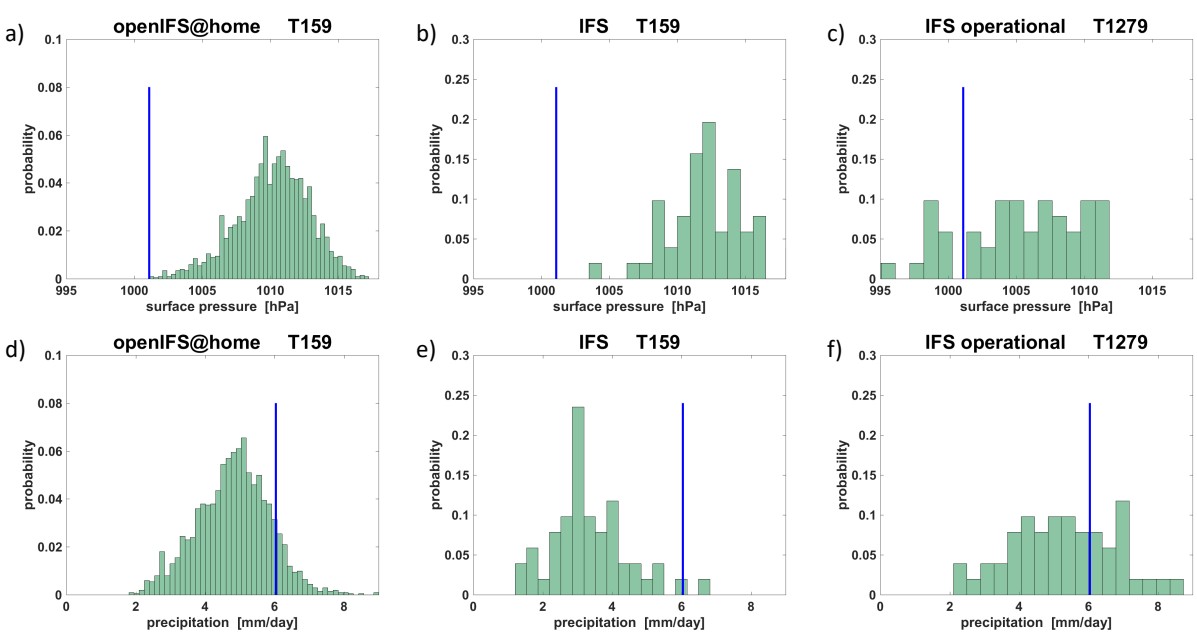

**Figure 4: Ensemble forecast distribution in Northern Scotland of mean sea-level pressure (first row) and total precipitation (second row) in the OpenIFS@home ensemble (left), in the IFS experiment (middle) and in the operational forecast (right). The**
**blue vertical line indicates the verification as derived from ECMWF's analysis. Mean sea-level pressure data are for 60h forecast lead time. Total precipitation data are accumulated between forecast lead times 60h-72h.**

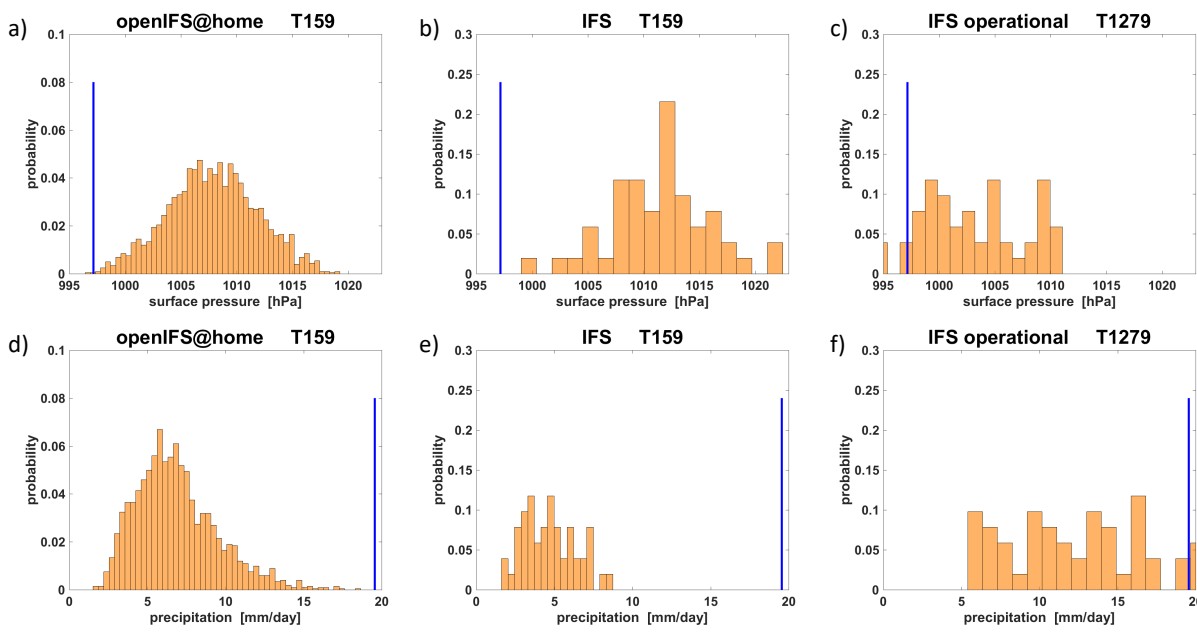

**Figure 5: Ensemble forecast distribution near Bergen of mean sea-level pressure (first row) and total precipitation (second row) in the OpenIFS@home ensemble (left), in the IFS experiment (middle) and in the operational forecast (right). The blue vertical line indicates the verification as derived from ECMWF's analysis. Mean sea-level pressure data are for 72h forecast lead time. Total precipitation data are accumulated between forecast lead times 60h-72h.**
