# Peer review of "OpenIFS@home version 1: a citizen science project for ensemble weather and climate forecasting"

_Geoscientific Model Development, 2020_

## Referee Comment (RC1) · Anonymous Referee #1 · 1 Nov 2020

The paper on OpenIFS@home is an interesting proof of concept paper. The paper neither goes into technical nor in scientific detail of the concept, but shows that producing coarse resolution ensembles OpenIFS on personal computers of citizens is possible and seems scientifically reasonable. As a model description paper for GMD I consider it too concise though.

This citizen science approach is limited to the use of external resources. The paper does not discuss how the citizens get involved, what there incentives are and whether there is feedback between the scientific community and the citizen science community.

However, I do applaud the approach to use OpenIFS on personal computers and in this way facilitate the generation of very large initialized ensembles. The application of OpenIFS is particular compared to earlier projects (climateprediction.net and

weather@home) in that it uses a code that is very close to the world-leading operational medium range prediction system. So, I think it is valuable to publish this and this specific application could be highlighted more.

The presented solution has strong limitations as well. The use of only one core limits the applicability to coarse resolution models due to memory constraints. While this is understandable it should be outlined clearly that this set up is only suitable for a limited set of scientific questions related to weather and climate issues. One also wonders whether this can be extended, given that multicore personal computers with different chips are omnipresent. It would be good have a discussion on the future scope related to this.

The scientific analysis is very limited. It is valuable to see some diagnostics for the 3 systems, coarse resolution with small and high number of ensembles and otherwise high resolution with small number of ensembles. The results show the limited applicability. Clearly resolution matters for the meteorological variables chosen and hence it is very hard to interpret the reliability of the 2000 member ensemble, although it is definitely an improvement compared to the coarse resolution, small ensemble.

The paper needs a bit more flesh in either science or technical implementation, and I think given the scope of GMD and the fact it is submitted as a model description paper it should be the latter. I would like to see more detail on the actual implementation of the software and the dependencies on the hardware and future development. So this needs more discussion on issues as efficiency, scalability, latency, energy use etc etc, so the performance of the code. This paper should include a more detailed comparison with climateprediction.net and weather@home. These could serve as a relevant benchmark, otherwise this paper is merely an introduction to a system. It would be really valuable to the scientific community to see how different, or similar, these applications are. Already in the abstract the innovation should be highlighted.

So my major comments are: 1) The paper needs to include more technical details and

diagnostics of this specific application of weather code in conjunction with BOINC and the application on personal computers. Currently, only Figure 2 gives some information on the timing, but there is no comparison to other systems (see also comment #2). There is hardly information on the performance of this particular code on the PCs (speed, energy efficiency, etc etc). 2) A benchmark misses, while there are good candidates. It would be good to compare the application with other known projects, such as weather@home and climateprediction.net 3) In the discussion extend the discussion to options and limitations for scientific use, both in terms of the type of questions that can and cannot be answered and the limits of not having the code itself. For instance, to what extend can a model with T159 horizontal resolution simulate cyclones and the extratropical transitions that such a cyclone goes through. The author show 'the power of large ensembles', but for this particular ensemble the resolution may be just as important. That is, a discussion on structural uncertainty misses and should be included as part of the discussion for what types of scientific questions can be addressed. The event chosen is in this regard interesting. 4) Related to the scientific use in comment #3, the statements on the simulated distributions, and in particular whether an event is part of the distribution, are not addressed with sufficient rigor. I suggest to perform a fit to the distributions, and a more in depth assessment of the probability of the event given each fitted distribution and whether one can actually state whether the distributions are different and whether the sampling is sufficient (the authors seem to indicate that the smaller ensemble is too small for that). Also, what is the impact of two different cycles of IFS? This likely affects the interpretation. 5) Extend the citizen science discussion. This a very limited approach to citizen science, only through using resources. What do the citizens get back from it? Is there interaction between the scientist and the citizen that leads to more insight in meteorology or more general in insight in science and social acceptance of science? Already in the introduction the authors state that the public awareness is raised, but is there proof of this?

Smaller remarks: The paper needs some more references. Already in the first paragraph of the introduction several projects are mentioned with no reference.

Citizen science and open science are strongly related. GMD is an open access journal and aims to have its data and code open. In this regards it is disappointing that the code has a license as it has. This is more a remark and the authors can hardly be blamed for this.

It would be good if more consistent use of persistent identifiers were used in the references of manuals and data. For instance for ECMWFs documentation, for the data that is stored at CEDA where the paper currently puts weblinks and email addresses which tend to break down quickly.
* * *

---

## Referee Comment (RC2) · Thomas M. Hamill (Referee) · 30 Dec 2020

This article is a straightforward description of the OpenIFS@home version 1 system that leverages an older version of ECMWF's forecast model at reduced resolution. My comments are minor.

1. Line 77: is an intended goal to contribute to understanding of the IFS system? Perhaps not, given the model version is older, but if so, mention here.

2. Line 169 and line 268. Which is it, 6 days or 8 days? Correct one or the other.

3. Line 179: So there are only 250 unique initial conditions, but 2000 forecasts by virtue of different stochastic noise via SPPT, correct? Some clarification here would be helpful.

[Figure]

4. Lines 224-228. Since in Figs. 4-5 you title panels with "IFS@ECMWF" and "Operational" it would be useful to the reader to clearly define these here. It's implied, but not explicit.

5. Line 282: I'm intrigued by the possibility that the large ensemble might be used for studying potential multi-modality. This seems to occur relatively frequently in small ensembles, but may be sampling variability. How often this persists with large ensembles would be useful to determine. Is multi-modality commonly real, or a sampling artifact?

6. Line 288: Any discussion of an update strategy? Eventually the model will lag so far behind ECMWF operations as to become less relevant as a tool.

7. Figure 1: this might be easier to see and interpret on a different map projection such as Lambert Conic Conformal.

Tom Hamill, NOAA PSL

---

## Author Comment (AC1) · 27 Jan 2021

Please see attached aggregated response to reviews

Please also note the supplement to this comment:
https://gmd.copernicus.org/preprints/gmd-2020-217/gmd-2020-217-AC1-supplement.pdf
* * *

---

## Author Response (AR1)

The paper on OpenIFS@home is an interesting proof of concept paper. The paper neither goes into technical nor in scientific detail of the concept, but shows that producing coarse resolution ensembles OpenIFS on personal computers of citizens is possible and seems scientifically reasonable. As a model description paper for GMD I consider it too concise though.

This citizen science approach is limited to the use of external resources. The paper does not discuss how the citizens get involved, what there incentives are and whether there is feedback between the scientific community and the citizen science community.

However, I do applaud the approach to use OpenIFS on personal computers and in this way facilitate the generation of very large initialized ensembles. The applica- tion of OpenIFS is particular compared to earlier projects (climateprediction.net and weather@home) in that it uses a code that is very close to the world-leading opera- tional medium range prediction system. So, I think it is valuable to publish this and this specific application could be highlighted more.

The presented solution has strong limitations as well. The use of only one core limits the applicability to coarse resolution models due to memory constraints. While this is understandable it should be outlined clearly that this set up is only suitable for a limited set of scientific questions related to weather and climate issues. One also wonders whether this can be extended, given that multicore personal computers with different chips are omnipresent. It would be good have a discussion on the future scope related to this.

The scientific analysis is very limited. It is valuable to see some diagnostics for the 3 systems, coarse resolution with small and high number of ensembles and otherwise high resolution with small number of ensembles. The results show the limited appli- cability. Clearly resolution matters for the meteorological variables chosen and hence it is very hard to interpret the reliability of the 2000 member ensemble, although it is definitely an improvement compared to the coarse resolution, small ensemble.

The paper needs a bit more flesh in either science or technical implementation, and I think given the scope of GMD and the fact it is submitted as a model description paper it should be the latter. I would like to see more detail on the actual implementation of the software and the dependencies on the hardware and future development. So this needs more discussion on issues as efficiency, scalability, latency, energy use etc etc, so the performance of the code. This paper should include a more detailed comparison with climateprediction.net and weather@home. These could serve as a relevant benchmark, otherwise this paper is merely an introduction to a system. It would be really valuable to the scientific community to see how

different, or similar, these applications are. Already in the abstract the innovation should be highlighted.

We thank the reviewer for their comments which we believe has led to improvement of the manuscript. Please find our detailed responses below.

So my major comments are:

1) The paper needs to include more technical details and diagnostics of this specific application of weather code in conjunction with BOINC and the application on personal computers. Currently, only Figure 2 gives some information on the timing, but there is no comparison to other systems (see also comment #2). There is hardly information on the performance of this particular code on the PCs (speed, energy efficiency, etc etc).

We have included a new subpanel to figure 2 (see below) which uses a single representative batch to compare the run times of OpenIFS@home for a model day of integration with the other existing models on the CPDN platform as an indicator. This shows that OpenIFS@home, a global model, is among the faster running models on the platform and comparable in run time to the different UK Met Office weather@home regions. It is worth noting however that even for the same application configurations, run times for an ensemble will vary depending on the specific mix of home computers that are running them and individual setup of project connectivity as explained in section 4.3.1 and so timings presented are only indicative rather than definitive. We have clarified the discussion of the timings presented in figure 2 in section 4.3.1 and included some additional metrics on the storage (3.2Gb) and memory (5.37 Gb RAM) requirements of OpenIFS@home running at this resolution. Additionally we have included a further reference on the design of running climate models on home computers under BOINC in climateprediction.net - Christensen et al, 2005.

[Figure]

Figure 2b: Run time information in hours per model day based on a representative batch for applications on the CPDN platform (note these numbers are indicative rather than definitive). ECMWF OpenIFS@home is depicted in yellow, UK Met Office weather@home (HadAM3P

with various HadRM3P  regions) configurations  in green (with light green indicating a 25km embedded region, mid green a 50km embedded region and dark green where only the global driving model is computed).  The UK Met Office low resolution coupled atmosphere-ocean model HadCM3 is shown in blue and the high resolution global atmosphere HadAM4 at N144 (~90km mid-latitudes) and  N216 (~60km mid-latitudes) are shown in purple.

2) A benchmark misses, while there are good candi- dates. It would be good to compare the application with other known projects, such as weather@home and climateprediction.net

We thank the reviewer for their comments and have included a comparison  to run times of other CPDN applications in the new figure  2b. It is worth noting that there are a number of different models and configurations on climateprediction.net and weather@home which have different run times depending on model resolution, embedded region, whether a coupled ocean is included etc.  Also these are primarily used for addressing different questions than those that will be addressed with OpenIFS@home.  The relevant aspect for  OpenIFS@home is whether this application will fit comfortably on a single machine (so can be run by volunteers) and what resolutions are possible in this framework as this  is the first spectral model that CPDN has deployed.  As can be seen from our response to your first major point we have included additional benchmark metrics.

3) In the discussion extend the discussion to options and limitations for scientific use, both in terms of the type of questions that can and cannot be answered and the limits of not having the code itself. For instance, to what extend can a model with T159 horizontal resolution simulate cyclones and the extratropical transitions that such a cyclone goes through. The author show 'the power of large ensembles', but for this particular ensemble the resolution may be just as important. That is, a discussion on structural uncertainty misses and should be included as part of the discussion for what types of scientific questions can be addressed. The event chosen is in this regard interesting.

We thank the reviewer for their comments and have included additional discussion on the key ingredients for a successful prediction of extreme events (realistic physical model, sufficient resolution, large ensembles) at the end of the meteorological results section (4.3.2) as follows:

"The forecast accuracy of the extreme meteorological conditions of storm Karl is influenced by 3 key factors: i) a good physical model that can simulate the atmospheric flow in highly baroclinic extratropical conditions as an extratropical low-pressure system; ii) higher horizontal resolution allows a better resolution of the storm, and iii) a large ensemble that samples a wide range of uncertainties given the simulated flow for a given resolution. OpenIFS@home is built on the world-leading NWP forecast model of ECMWF which enables our distributed forecasting system to use the most advanced science of weather prediction. Arguably, a storm like Karl will be better resolved with higher horizontal resolution, as becomes clear in our demonstration of the IFS performance in two contrasting resolutions. However, there are other meteorological phenomena where horizontal resolution does not play a similarly large role in the successful prediction of extreme events. For these situations, the availability of very large ensembles that enable a meaningful sampling of the tails of the distribution and with it the risks for extreme outcomes, will be most valuable. Our storm Karl analysis has made that point very clear by showing a substantial improvement in the probabilistic forecasts of both very low surface pressure (and associated winds) and large rainfall totals."

We have further added a paragraph at the very end on possible future uses of the system:

"In terms of potential areas of future scientific use of openIFS@home, research on understanding and predicting compound extreme events (for example, a heat wave in conjunction with a meteorological and hydrological drought) will be of interest. ECMWF's operation ensemble size of 51 members makes such investigations very difficult and limited in their scope, while the very large ensemble set-up of openIFS@home provides an ideal framework for the required sample sizes of multi-variate studies. We are planning to use the system for predictability research on a range of time scales from days to weeks and months, with potential idealised climate applications also feasible in the longer term."

Our aim in the future is to use the system for seasonal forecast simulations where the resolution used here is closer to standard operational resolutions.  In the experimental design process, horizontal resolution is just one factor as physical processes, parameterisations and vertical resolution also need to  be considered.   This is not a unique limitation for OpenIFS@home and is also a consideration when designing experiments for HPC or other types of infrastructure.

On code availability, users are able to access the model code if they apply for a licence from ECMWF and OpenIFS@home would allow custom versions of OpenIFS  as long as it passes suitable quality control and assurance.

4) Related to the scientific use in comment #3, the statements on the simulated distributions, and in particular whether an event is part of the distribution, are not addressed with sufficient rigor. I suggest to perform a fit to the distributions, and a more in depth assessment of the probability of the event given each fitted distribution and whether one can actually state whether the distributions are different and whether the sampling is sufficient (the authors seem to indicate that the smaller ensemble is too small for that). Also, what is the impact of two different cycles of IFS? This likely affects the interpretation.

This paper mainly aims to introduce the set-up of OpenIFS@home as a first implementation of the ECMWF operational NWP model in the BOINC framework. It demonstrates what can be done with such a set-up but does not aim to provide a full scientific study. This is a single case study as proof of concept and the different distributions can be clearly seen in the histograms provided. The histograms for our case study are so clear-cut that we are not convinced a distributional fit would add any further understanding (instead, it would introduce assumptions on the underlying distribution and large uncertainties for the smaller ensemble size). The effect of resolutions vs ensemble size has been discussed explicitly in section 4.3.2

5) Extend the citizen science dis- cussion. This a very limited approach to citizen science, only through using resources. What do the citizens get back from it? Is there interaction between the scientist and the citizen that leads to more insight in meteorology or more general in insight in science and social acceptance of science? Already in the introduction the authors state that the public awareness is raised, but is there proof of this?

There are indeed many forms of citizen science ranging from the more passive form (in this example) through to those that require more active participation. The purpose of this paper  is to illustrate the new modelling capability within the climateprediction.net platform, rather than the citizen science aspects of the platform itself.   As detailed in the original

climateprediction.net (Christensen et al, 2005) and BOINC papers (Anderson, 2004) volunteers are able to engage directly with scientists via the project forums and message boards where news items are circulated about simulations and the questions they are trying to address. Additionally climateprediction.net has recently implemented a new feature which notifies users of the scientific output that they have contributed towards. This is also on top of numerous news and media items on the results found by project scientists (which often would not be possible without the large ensembles generated by public computing) or experiments that are undertaken in conjunction with specific media outlets (e.g. The Guardian, BBC). We have made the following alterations to the manuscript (at L51) to reflect this:

"Volunteers can sign-up to CPDN through the project website and are engaged and retained through the mechanisms detailed in (Christensen et al., 2005). As well as facilitating large ensemble climate simulations the project has also increased public awareness of climate change related issues. Through the CPDN platform, volunteers are notified of the scientific output that they have contributed towards (complete with links to the academic publication) and through the project forums and message boards can engage directly with scientists about the experiments being undertaken. Public awareness is also raised by press coverage of the project (e.g. Gadgets that give back: awesome eco-innovations, from Turing Trust computers to the first sustainable phone; Climate Now | Five ways you can become a citizen scientist and help save the planet), scientific outputs (e.g. 'weather@home' offers precise new insights into climate change in the West; How your computer could reveal what's driving record rain and heat in Australia and NZ; Looking, quickly, for the fingerprints of climate change) and through live experiments undertaken directly with media outlets such as The Guardian (Schaller et al., 2016) and British Broadcasting Corporation (BBC, Rowlands et al. 2012)."

Smaller remarks: The paper needs some more references. Already in the first para- graph of the introduction several projects are mentioned with no reference.

Additional references have been included in the introduction, in particular for Garden BirdWatch and Zooniverse.

Citizen science and open science are strongly related. GMD is an open access journal and aims to have its data and code open. In this regards it is disappointing that the code has a license as it has. This is more a remark and the authors can hardly be blamed for this.

Noted. We argue that enabling use of OpenIFS on this platform (under the binary licence mentioned) aids open science by providing another route by which researchers (particularly those in developing countries) can get supported access to use OpenIFS and form closer links with the scientists involved. The driving code between BOINC and the OpenIFS binary is provided open-source (Bowery and Carver, 2020, Zenodo) and we expect that (based on CPDN experience with Met Office models) that only rarely scientists expect to make modifications to the actual source code of the model itself but tend to use the model as-is.

It would be good if more consistent use of persistent identifiers were used in the ref- erences of manuals and data. For instance for ECMWFs documentation, for the data that is stored at CEDA where the paper currently puts weblinks and email addresses which tend to break down quickly.

We are corresponding with CEDA to coordinate the data deposit. When this is complete, the data will be issued with a persistent identifier which will be referenced in this paper. ECMWF are issuing dois for documentation which will also be referenced in this paper.

This article is a straightforward description of the OpenIFS@home version 1 system that leverages an older version of ECMWF's forecast model at reduced resolution. My comments are minor.

We thank Tom Hamill for his review and include specific replies to his comments below.

1. Line 77: is an intended goal to contribute to understanding of the IFS system? Perhaps not, given the model version is older, but if so, mention here.

This is not really an intended goal for us so we have left this as is.

2. Line 169 and line 268. Which is it, 6 days or 8 days? Correct one or the other.

Thank you for pointing out this inconsistency. This has now been corrected to 6 days.

3. Line 179: So there are only 250 unique initial conditions, but 2000 forecasts by virtue of different stochastic noise via SPPT, correct? Some clarification here would be helpful.

We have clarified the text around the initial conditions as follows:

"The ECMWF data assimilation system was used to create 250 perturbed initial states. Each of these 250 states was then used for 8 forecasts in the 2,000 member ensemble. A different forecast realisation for each set of 8 forecasts was generated by enabling the stochastic noise in the OpenIFS physical parametrizations. "

4. Lines 224-228. Since in Figs. 4-5 you title panels with "IFS@ECMWF" and "Opera- tional" it would be useful to the reader to clearly define these here. It's implied, but not explicit.

Agreed and we have updated figures 4 and 5 accordingly. We have added the horizontal resolution directly in the titles of the plots which will make a contrast of the resolutions much clearer.

5. Line 282: I'm intrigued by the possibility that the large ensemble might be used for studying potential multi-modality. This seems to occur relatively frequently in small en- sembles, but may be sampling variability. How often this persists with large ensembles would be useful to determine. Is multi-modality commonly real, or a sampling artifact?

We very much agree with this point. As can already be seen in our analysis of storm Karl in Figures 4 and 5, most of the perceived "jumpiness" in the forecast distribution based on 51 ensemble members has been smoothed out with the large ensemble. Storm Karl is not ideal to study multi-modality because of its strong zonal flow component, but other applications where the effects of sampling variability vs true multi-modality will hopefully be studied in the future.

6. Line 288: Any discussion of an update strategy? Eventually the model will lag so far behind ECMWF operations as to become less relevant as a tool.

Future updates to OpenIFS@home will include support for more recent versions of the application. We have added a comment to this effect in the conclusions.

7. Figure 1: this might be easier to see and interpret on a different map projection such as Lambert Conic Conformal.

We have decided to keep the current projection as this is aimed at the non-specialist. However we have updated figure 1 for clarity.

Tom Hamill, NOAA PSL